# CoDiffSplat: Sparse-View Generalizable 3D Gaussian Splatting with Single-Step Conditional Diffusion

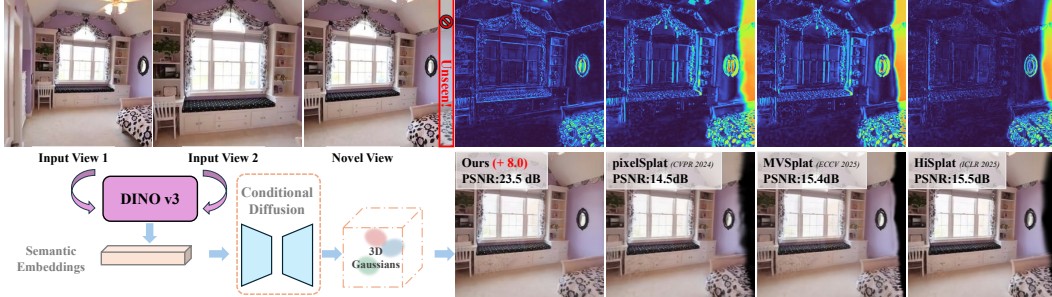

Figure 1: **Comparison between CoDiffSplat and baselines.** Previous methods are pixel-aligned and depth-based, often failing to recover geometry in ambiguous or occluded regions (e.g., unseen surfaces). In contrast, CoDiffSplat leverages semantic embeddings and conditional diffusion to generate refined 3D Gaussians, enabling high-fidelity reconstruction of uncertain areas. Residual maps between rendered results and ground truth are shown in the top-right corner.

## Abstract

Generalizable 3D Gaussian Splatting (G-3DGS) has emerged as a promising approach for novel view synthesis under sparse-view settings. However, existing frameworks remain restricted by pixel-aligned Gaussian estimation, which struggles in regions that are partially observed or occluded, often resulting in incomplete geometry and structural collapse. To overcome these challenges, we propose CoDiffSplat, a new framework that couples semantic-conditioned latent diffusion with 3D Gaussian splatting. Our design departs from conventional diffusion applied on image feature maps: instead, a lightweight single-step diffusion directly refines Gaussian parameters, ensuring efficiency while preserving geometric consistency. In addition, we introduce a Cross-View Entropy-Aware (CEA) module that aggregates multi-view semantics and geometry into robust conditional embeddings, enabling diffusion to resolve ambiguities under occlusion and sparse overlap. Comprehensive experiments on multiple benchmarks demonstrate that CoDiffSplat consistently improves geometric quality and structural completeness, especially under challenging extrapolation settings. Our study establishes conditional diffusion as a scalable refinement mechanism for sparse-view 3D reconstruction, advancing the reliability of generalizable Gaussian splatting.

## 1 Introduction

3D reconstruction is a cornerstone of computer vision, powering applications such as autonomous driving, virtual reality, and augmented reality. Recently, 3D Gaussian Splatting (3DGS) Kerbl et al. (2023) has emerged as a powerful paradigm for scene representation and novel view synthesis (NVS) Buehler et al. (2001). By modeling a scene as a mixture of 3D Gaussians and leveraging differentiable rasterization, 3DGS enables high-fidelity and real-time rendering from dense multi-view images. However, conventional 3DGS methods typically rely on per-scene optimization with

dense input views, which limits scalability and requires costly data acquisition. To alleviate these constraints, generalizable 3DGS (G-3DGS) has been developed to reconstruct scenes from only a few input views. These approaches employ pre-trained feed-forward models Chen et al. (2025b); Charatan et al. (2024); Zhang et al. (2025); Wewer et al. (2025) that encode scene priors from large-scale datasets, enabling rapid inference without scene-specific optimization.

Despite this progress, existing G-3DGS frameworks remain fundamentally restricted by their strong reliance on pixel-aligned unprojection Charatan et al. (2024); Chen et al. (2025b). Specifically, each pixel is mapped to a fixed number of Gaussian primitives based on estimated depth, making reconstruction highly sensitive to depth errors. However, under sparse-view conditions, depth estimation often suffers from occlusions, weak textures, and limited viewpoint overlap. Moreover, rigid pixel-level alignment hinders the recovery of unobserved or partially visible regions, frequently producing 'black holes' or collapsed structures in the final rendering (Fig. 1).

To address these limitations, we propose to **shift the reconstruction paradigm from pixel-space alignment to refinement in the Gaussian domain.** While previous works attempted to address uncertainty through depth regularization or feature fusion, they remain inherently constrained by pixel-level priors. Motivated by the remarkable success of diffusion models in 3D content generation tasks (such as text-to-3D synthesis Lin et al. (2025); He et al. (2025); Cao et al. (2024)), we investigate whether diffusion can synthesize missing structures and compensate for uncertainty. Building on this intuition, we introduce **CoDiffSplat**, a novel G-3DGS framework that integrates conditional diffusion to refine initial pixel-aligned Gaussians into geometrically consistent and complete structures. Unlike conventional diffusion pipelines, which require costly iterative denoising, we show that *a single-step refinement suffices* to correct geometric inconsistencies, turning diffusion into an efficient correction module rather than a full generative process. Consequently, our pipeline both *retains the speed advantages of feed-forward G-3DGS approaches and alleviates the dependency on perfect depth alignment*. However, a key challenge in NVS is the absence of explicit text prompts. While pseudo-captioning (e.g., BLIP Li et al. (2023)) offers weak supervision, it often overlooks fine-grained details Patni et al. (2024). To address this, we design a **Cross-View Entropy-Aware (CEA)** module that fuses multi-view semantic cues with geometry-uncertainty signals, yielding detail-preserving embeddings that guide the diffusion process toward challenging regions. Technically, to mitigate the training difficulty and high computational cost associated with diffusion, CoDiffSplat performs denoising in latent Gaussian space using a lightweight DiT backbone.

The main contributions are summarized as follows:

- We formulate sparse-view G-3DGS as a *latent Gaussian refinement* problem and present CoDiffSplat, which employs a lightweight DiT-based diffusion model with a single-step denoiser to relax rigid pixel alignment and restore missing geometry.

- We propose the CEA module, which combines semantic cues with depth-distribution entropy to emphasize uncertain regions and provide fine-grained conditional guidance for diffusion.

- We validate CoDiffSplat on standard benchmarks across interpolation and extrapolation settings, and CoDiffSplat consistently improves fidelity in both settings. In particular, relative to the SOTA HiSplat Chen et al. (2025b), it achieves a **+2.32 dB** PSNR gain on RealEstate10K Zhou et al. (2018) in extrapolated views, while maintaining competitive computational efficiency.

## 2 RELATED WORK

### 2.1 NOVEL VIEW SYNTHESIS

Novel view synthesis (NVS) aims to render photo-realistic images from novel viewpoints using only a limited set of input images Buehler et al. (2001). Neural Radiance Fields (NeRF) Mildenhall et al. (2020); Yu et al. (2021); Pumarola et al. (2021); Barron et al. (2021; 2022) models scenes as continuous volumetric radiance fields parameterized by neural networks. While NeRF-based methods have yielded impressive results in dense multi-view settings, they typically suffer from slow training times, high memory usage, and suboptimal performance under sparse viewpoints due to their heavy reliance on per-ray MLP evaluations. In contrast, 3D Gaussian Splatting (3DGS) Kerbl et al.

(2023); Yu et al. (2024); Yang et al. (2024) introduces an explicit scene representation by modeling surfaces with anisotropic 3D Gaussian primitives. Each Gaussian is described by its position, covariance, color, and opacity, which can be differentiably rendered via a forward projection to the image plane. This explicit design significantly accelerates the rendering process compared to vanilla NeRF pipelines, yet many existing 3DGS approaches still assume relatively dense coverage of views for accurate geometry and appearance reconstruction. Consequently, their performance deteriorates for extremely sparse inputs, where geometric ambiguity and insufficient texture cues become major challenges.

## 2.2 SPARSE-VIEW GENERALIZABLE 3DGS

Sparse-View generalizable 3DGS methods focus on learning a feed-forward model capable of handling unseen scenes without per-scene re-optimization. PixelSplat Charatan et al. (2024) predict 3D Gaussian parameters from sparse multi-view inputs, leveraging an epipolar transformer for depth estimation. MVSplat Chen et al. (2025b) relies on cost-volume construction via plane sweeping to infer depth distributions. TranSplat Zhang et al. (2025) introduces a transformer-based architecture with depth-aware deformable matching for coarse-to-fine refinement. HiSplat Tang et al. (2025b) integrates hierarchical Gaussian features, leveraging iterative Gaussian alignment. eFreeSplat Min et al. (2024) eliminates epipolar priors by leveraging cross-view completion. DepthSplat Xu et al. (2025) bridges Gaussian splatting and depth estimation by leveraging pre-trained monocular depth features to enhance multi-view depth prediction. Despite these diverse approaches, most still rely heavily on estimated depth maps, which can become noisy or unreliable under sparse-view conditions. They also often utilize pixel-aligned Gaussian estimation, causing difficulties in recovering fine details or resolving ambiguities in unseen regions.

## 2.3 3D-AWARE DIFFUSION METHODS

Recent diffusion-based methods have expanded from 2D image generation to 3D-aware tasks such as image-to-3D and text-to-3D generation Lin et al. (2025); Cao et al. (2024); Nichol et al. (2022); Hong et al. (2024); Li et al. (2024); Tang et al. (2025a); Shi et al. (2024); Yang et al. (2025). Image-to-3D approaches attempt to recover 3D content from a single image, but they often suffer from ambiguity and incomplete geometry due to the lack of multi-view constraints. Text-to-3D pipelines leverage diffusion priors for novel asset creation, and can be adapted to image inputs through captioning, thereby unifying the two paradigms. While diffusion provides strong generative flexibility, existing 3D-aware frameworks typically lack explicit multi-view consistency and assume access to dense supervision, which limits their applicability in sparse-view reconstruction. These limitations motivate our work, where we integrate diffusion-based priors with Gaussian splatting to enhance geometric completeness and robustness under sparse and extrapolated views.

## 3 METHODOLOGY

Following the G-3DGS framework Charatan et al. (2024); Chen et al. (2025b); Zhang et al. (2025); Min et al. (2024); Xu et al. (2025); Tang et al. (2025b), the input consists of $V$ sparse-view images $\mathcal{I} = \{I_1, I_2, \ldots, I_V\}$, where each image $I_i \in \mathbb{R}^{H \times W \times 3}$ is accompanied by its camera projection matrix derived from intrinsic and extrinsic parameters. The goal is to reconstruct the underlying 3D scene as a set of Gaussian primitives $\Theta = \{G_j\}_{j=1}^N$, where each primitive $G_j$ is parameterized by its center $\boldsymbol{\mu}_j$, opacity $\alpha_j$, covariance $\boldsymbol{\Sigma}_j$, and color $\boldsymbol{c}_j$. The number of Gaussians is typically set to $N = H \times W \times V$, corresponding to the input resolution and number of views. These primitives are subsequently rendered into novel views through differentiable Gaussian splatting Kerbl et al. (2023).

As illustrated in Figure 2, CoDiffSplat adopt a latent conditional diffusion paradigm tailored for sparse-view reconstruction. Specifically, our pipeline comprises two branches: one employ a multi-view Gaussian encoder to generate a coarse initialization of latent Gaussian parameters, and another for extracting a Cross-view Entropy-Aware (CEA) embedding as condition of subsequent diffusion model. Instead of operating over image-space noise or raw latent feature maps, our diffusion backbone denoises in the latent space of Gaussian parameters, refining coarse geometry and appearance under semantic guidance. The refined latent Gaussians are decoded via an upsampling decoder to produce full-resolution Gaussians.

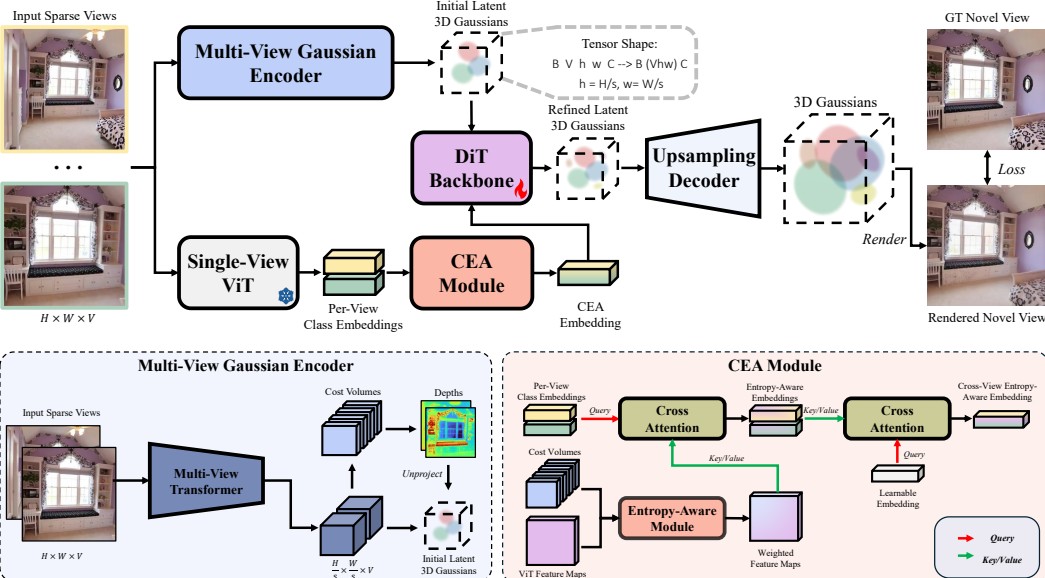

Figure 2: **Overview of CoDiffSplat.** Given sparse input views, our framework first constructs an initial set of latent 3D Gaussians through the multi-view Gaussian encoder. In parallel, a frozen single-view ViT extracts per-view class embeddings, which are fused by the proposed Cross-view Entropy-Aware (CEA) module to produce a unified multi-view semantic embedding. This embedding conditions the DiT backbone to refine the latent Gaussians, followed by an upsampling decoder that restores them to the original spatial resolution. The refined 3D Gaussians are rendered to synthesize novel views, and the entire model is optimized end-to-end using photometric loss.

### 3.1 GAUSSIAN INITIALIZATION

A reliable initialization is essential for stable diffusion-based refinement. We therefore introduce a multi-view Gaussian encoder that constructs latent 3D Gaussians directly from sparse input views. By aggregating multi-view features and inferring coarse scene geometry, this module produces a structured and pixel-aligned Gaussian representation. These initialized Gaussians provide strong geometric priors, serving as an effective starting point for the subsequent diffusion process.

#### 3.1.1 LATENT FEATURE EXTRACTION

Diffusion models typically require many time steps to converge. Recently, Stable Diffusion Rombach et al. (2022) introduced an approach that shifts the denoising process into a learned latent representation, significantly easing optimization. Inspired by this paradigm, we employ a CNN and Transformer Vaswani et al. (2017) architecture to project multi-view images into latent space. Specifically, each input image $I^i$ is first passed through a shallow ResNet He et al. (2016) to produce $s\times$ downsampled feature maps. To efficienctly integrates cross-view information, we then leverage a multi-view Swin Transformer Liu et al. (2021b) which contains self- and cross-attention layers to obtain multi-view-aware features $\boldsymbol{F}^i \in \mathbb{R}^{\frac{H}{s} \times \frac{W}{s} \times C}$, where $C$ is the feature dimension. This remains computationally tractable while preserving cross-view interactions. We then utilize the multi-view features to initialize latent Gaussian parameters.

#### 3.1.2 COARSE MATCHING

To establish a reliable geometric prior for the subsequent diffusion process, we construct cost volumes with plane sweeping Xu et al. (2023); Yao et al. (2018) to model multi-view feature correspondences, which facilitates the initialization of coarse Gaussian parameters. Specifically, for each view $i$, we uniformly sample $D$ depth candidates $\{d_m\}_{m=1}^{D}$ in the inverse depth domain between near and far planes. Features from other views ($\boldsymbol{F}^j, j \neq i$) are warped to view $i$ using camera parameters and depth candidate $d_m$, producing $D$ warped features $\{\boldsymbol{F}_{d_m}^{j \to i}\}_{m=1}^{D}$. The correlation $\boldsymbol{C}_{d_m}^i$ between $\boldsymbol{F}^i$

and $\boldsymbol{F}_{d_m}^{j \to i}$ is computed with the dot-product operation:

$$C_{\text{dim}}^i = \frac{F_i \cdot F_{d_m}^{j \to i}}{\sqrt{C}}, \quad m = 1, 2, \dots, D. \tag{1}$$

We average $\boldsymbol{C}_{d_m}^i$ across all other views to form the cost volume $\mathbf{C}^i \in \mathbb{R}^{\frac{H}{s} \times \frac{W}{s} \times D}$. Subsequently, we apply the softmax operation to compute per-view depth map:

$$\boldsymbol{Z}^i = \text{softmax}(\mathbf{C}^i) \cdot \boldsymbol{\Lambda}, \tag{2}$$

where $\boldsymbol{\Lambda} = [d_1, d_2, \dots, d_D]$ are the depth candidates. The coarse depth map $\boldsymbol{Z}^i \in \mathbb{R}^{\frac{H}{s} \times \frac{W}{s}}$ is then unprojected to form preliminary Gaussian centers $\boldsymbol{\mu}$ using the camera parameters, and other Gaussian parameters are predicted by additional lightweight heads from feature maps. This ensures that the initial positions are geometrically consistent with the input views. The constructed cost volumes $\mathbf{C}^i$ are further retained as conditional inputs to the CEA module, providing pixel-wise structural cues about the scene.

## 3.2 CONDITIONAL DIFFUSION

While recent works have applied diffusion to 3D reconstruction Chen et al. (2025a); Wu et al. (2025; 2024), sparse-view NVS suffers from a critical limitation: the generative prior lacks sufficient constraints, resulting in unstable refinements and hallucinated geometry. We address this challenge with a conditional diffusion module tailored for Gaussian splatting. Guided by cross-view entropy-aware (CEA) embeddings, the diffusion directly refines Gaussian parameters in latent space through a single-step process, ensuring both efficiency and consistency.

### 3.2.1 CROSS-VIEW ENTROPY-AWARE EMBEDDING EXTRACTION

Existing conditional embeddings, such as BLIP text features Li et al. (2023), mainly capture global semantics and tend to overemphasize large salient objects, suppressing fine-grained details and often causes boundary artifacts Patni et al. (2024). Furthermore, since these embeddings are extracted independently for each view, they contain redundant or overlapping content, which further amplifies occlusion-induced information loss. To address these issues, we introduce a Cross-View Entropy-Aware (CEA) embedding that highlights uncertain or weakly constrained regions while consolidating cross-view redundancy into a comprehensive representation.

We first propose an entropy-aware module that leverages the cost volume $\mathbf{C}^i$ to compute the matching entropy $H^i$, thereby identifying weakly constrained regions (e.g., unseen, occluded, or textureless areas). A single-view ViT encoder pretrained with DINOv3 Siméoni et al. (2025) provides per-view class embeddings $\mathbf{E}_{CLS}^i$ as well as feature maps $\mathbf{F}^i$. At each pixel $p$ in view $i$, the depth posterior is estimated by applying a softmax along the depth axis of the cost volume:

$$P^i(d \mid p) = \text{softmax}_d\big(\mathbf{C}^i(p, d)\big). \tag{3}$$

The matching entropy is then defined as:

$$H^i(p) = -\sum_{d \in \Lambda} P^i(d \mid p) \log P^i(d \mid p), \tag{4}$$

and further normalized into per-pixel weights $w^i(p)$. This entropy quantifies the degree of multi-view uncertainty, where larger values of $w^i(p)$ indicate more ambiguous or under-constrained regions. The feature map $\mathbf{F}^i$ is reweighted accordingly to obtain an entropy-aware representation:

$$\tilde{\mathbf{F}}^i(p) = w^i(p)\,\mathbf{F}^i(p). \tag{5}$$

Subsequently, the class embedding $\mathbf{E}_{CLS}^i$ is linearly projected into queries, while the weighted feature map $\tilde{\mathbf{F}}^i$ is projected into keys and values:

$$\mathbf{Q}^i = \mathbf{E}_{CLS}^i \mathbf{W}_Q, \quad \mathbf{K}^i = \tilde{\mathbf{F}}^i \mathbf{W}_K, \quad \mathbf{V}^i = \tilde{\mathbf{F}}^i \mathbf{W}_V. \tag{6}$$

A cross-attention mechanism then refines the class embedding into an entropy-aware embedding:

$$\tilde{\mathbf{E}}_{CLS}^i = \text{CrossAttn}\big(\mathbf{Q}^i, \mathbf{K}^i, \mathbf{V}^i\big) + \mathbf{E}^i, \tag{7}$$

which adaptively integrates fine-grained cues, especially from regions of high uncertainty.

To further eliminate inter-view redundancy and alleviate the masking effect of large salient objects, we employ a cross-view Perceiver-style attention mechanism with a set of learnable latent queries. Denote these learnable queries as $\mathbf{Q}_\ell$. By concatenating the refined per-view embeddings $\tilde{\mathbf{E}}^i$, we construct the keys and values $\mathbf{K}_{\mathrm{mv}}$ and $\mathbf{V}_{\mathrm{mv}}$, respectively, and compute the CEA embeddings through a cross-attention:

$$\mathbf{E}_{\mathrm{CEA}} = \mathrm{CrossAttn}\big(\mathbf{Q}_\ell, \mathbf{K}_{\mathrm{mv}}, \mathbf{V}_{\mathrm{mv}}\big). \tag{8}$$

The resulting representation $\mathbf{E}_{\mathrm{CEA}}$ aggregates cross-view semantics while suppressing redundant biases, thereby providing a stable and informative conditioning signal for the subsequent diffusion.

### 3.2.2 GAUSSIAN-STRUCTURED REPRESENTATION

Although Gaussian primitives are inherently unordered point sets, our initialization procedure imposes a pixel-to-Gaussian mapping, enabling the latent parameter tensor to be structured as $\Theta_l \in \mathbb{R}^{B \times V \times h \times w \times C}$, where $B$ denotes the batch size, $V$ the number of views, $h \times w = \frac{H}{s} \times \frac{W}{s}$ the latent spatial resolution, and $C$ the dimensionality of Gaussian parameters. This structured representation aligns with the pixel domain and facilitates efficient parameter organization. However, directly applying convolutional architectures such as UNet Ronneberger et al. (2015) to this tensor may introduce spurious grid-based inductive biases, which can potentially oversmooth Gaussian distributions and compromise geometric fidelity. To better respect the unordered nature of Gaussian primitives, we instead employ a transformer-based diffusion backbone. Specifically, we flatten the structured tensor into a sequence of tokens:

$$\Theta_l = \mathtt{rearrange}(\Theta_l, \mathtt{B\ V\ h\ w\ C} \to \mathtt{B\ (Vhw)\ C}), \tag{9}$$

and process it using a Diffusion Transformer (DiT) backbone Peebles & Xie (2023), which is well suited for irregular and unordered data.

### 3.2.3 SINGLE-STEP REFINEMENT

Motivated by recent findings that single-step diffusion suffices for refinement tasks Wu et al. (2025); Lin et al. (2024); Qu et al. (2025), we design the diffusion stage as a *residual correction module* rather than a full generative process. Specifically, the initialized latent Gaussians as $\Theta_l$ can be served as coarse but noisy approximations of the target distribution. The refinement is performed through a single-step correction:

$$\hat{\Theta}_l = \Theta_l + f_\theta(\Theta_l, \mathbf{E}_{\mathrm{CEA}}), \tag{10}$$

where $f_\theta$ is a DiT-based predictor conditioned on CEA embeddings $\mathbf{E}_{\mathrm{CEA}}$. This one-step formulation mitigates structured noise while preserving geometric consistency, avoiding the overcorrection and instability often observed in multi-step denoising Lin et al. (2024); Qu et al. (2025).

## 3.3 RENDERING AND TRAINING LOSS

Datasets for NVS do not provide explicit ground-truth 3DGS supervision, which prevents the use of a conventional forward–reverse diffusion process. In particular, there is no well-defined target domain for injecting noise, and thus no tractable formulation for noise prediction. We therefore adopt a straightforward latent estimation scheme that directly predicts the denoised Gaussian parameters. Consequently, the training objective reduces to a photometric reconstruction loss. The generated Gaussian parameters $\Theta$ are used to render novel views via 3DGS's differentiable rasterization Kerbl et al. (2023). The model is trained end-to-end using a photometric loss between rendered images $\mathcal{R}(\theta)$ and ground truth target views $\mathcal{I}_{\mathrm{gt}}$, combining $\ell_2$ and LPIPS Zhang et al. (2018b) terms:

$$\mathcal{L}_{\mathrm{photo}} = \|\mathcal{R}(\Theta) - \mathcal{I}_{\mathrm{gt}}\|_2^2 + 0.05 \cdot \mathrm{LPIPS}(\mathcal{R}(\Theta), \mathcal{I}_{\mathrm{gt}}). \tag{11}$$

## 4 EXPERIMENTS AND DISCUSSIONS

### 4.1 EXPERIMENTAL SETTINGS

We train and evaluate our approach on two large-scale datasets, RealEstate10K Zhou et al. (2018) and ACID Liu et al. (2021a). RealEstate10K contains home walkthrough videos from YouTube,

Table 1: **Comparison of interpolated NVS.** We evaluate performance on the RealEstate10K and ACID datasets by rendering three novel interpolation views from two reference viewpoints, averaging across all scenes. The dataset's training and testing split follows the identical protocol established by pixelSplat. Note that 3DGS-based methods render extremely fast ($\sim 500$FPS).

| Method | RealEstate10K | | | ACID | | | Inference Time |
| | PSNR↑ | SSIM↑ | LPIPS↓ | PSNR↑ | SSIM↑ | LPIPS↓ | (s) |
|---|---|---|---|---|---|---|---|
| pixelSplat Charatan et al. (2024) | 25.89 | 0.858 | 0.142 | 28.14 | 0.839 | 0.150 | 0.104 |
| MVSplat Chen et al. (2025b) | 26.39 | 0.869 | 0.128 | 28.25 | 0.843 | 0.144 | **0.044** |
| eFreeSplat Min et al. (2024) | 26.45 | 0.865 | 0.126 | 28.30 | 0.851 | 0.140 | 0.061 |
| TranSplat Zhang et al. (2025) | 26.69 | 0.875 | 0.125 | 28.35 | 0.845 | 0.143 | 0.087 |
| HiSplat Tang et al. (2025b) | 27.21 | 0.881 | 0.117 | 28.75 | 0.853 | **0.133** | 0.510 |
| **CoDiffSplat (Ours)** | **27.56** | **0.888** | **0.114** | **28.77** | **0.855** | **0.133** | 0.089 |

Table 2: **Comparison of extrapolated NVS on RealEstate10K.** We evaluate model performance on RealEstate10K by rendering three novel extrapolated views from two reference views, averaging across all scenes under identical training settings.

| Method | PSNR↑ | SSIM↑ | LPIPS↓ |
|---|---|---|---|
| pixelSplat Charatan et al. (2024) | 21.76 | 0.779 | 0.217 |
| MVSplat Chen et al. (2025b) | 21.92 | 0.787 | 0.199 |
| TranSplat Zhang et al. (2025) | 21.89 | 0.791 | 0.201 |
| HiSplat Tang et al. (2025b) | 22.01 | 0.794 | 0.191 |
| **CoDiffSplat (Ours)** | **24.33** (+2.32) | **0.846** (+0.052) | **0.152** (−0.039) |

comprising 67,477 training scenes and 7,289 testing scenes. ACID consists of aerial nature footage captured by drones, split into 11,075 training scenes and 1,972 testing scenes. Both datasets are calibrated via Structure-from-Motion (SfM) Schönberger & Frahm (2016), which provides per-frame camera intrinsics and extrinsics. Following prior works Charatan et al. (2024); Chen et al. (2025b), we use two context images as input and render three novel target views for each test scene. To assess the model's comprehensive understanding of 3D scenes, we evaluate not only conventional interpolated NVS but also *extrapolated* NVS, where target viewpoints lie beyond the reference range. We adopt the training curriculum from pixelSplat Charatan et al. (2024), increasing the sampling interval of target views up to 45 frames before and after the reference views to accommodate extrapolation. To evaluate visual fidelity, we compare the images rendered by each method with the corresponding ground truth frames by computing a peak signal-to-noise ratio (PSNR), structural similarity index (SSIM) Wang et al. (2004), and perceptual distance (LPIPS) Zhang et al. (2018a). Please refer to the Appendix A.1 for implementation details. We also conduct zero-shot cross-dataset generalization experiments following the MVSplat protocol, which are discussed in detail in Appendix A.2.1.

## 4.2 MAIN RESULTS

### 4.2.1 INTERPOLATED NOVEL VIEW SYNTHESIS

Table 1 shows that CoDiffSplat achieves better performance on both RealEstate10K and ACID. While prior G-3DGS approaches differ in architectural details, they all rely on pixel-aligned Gaussian parameter estimation in a purely feed-forward manner, which often leads to blurred details, geometric distortions, or failure in occluded regions (Figure 3). In contrast, our conditional diffusion leverages CEA embeddings that effectively fuse semantic and geometric cues across views, enabling accurate reconstruction of fine structures and hidden surfaces. Importantly, the single-step refinement introduces only marginal computational overhead (0.089s per frame), remaining competitive with lightweight baselines while being substantially faster than HiSplat (0.510s). These results demonstrate that CoDiffSplat achieves a favorable balance between efficiency and reconstruction fidelity, with enhanced generalization in both indoor and outdoor sparse-view settings. Additional qualitative comparisons on RealEstate10K are provided in Appendix A.2.5.

### 4.2.2 EXTRAPOLATED NOVEL VIEW SYNTHESIS

To further assess the generalization ability of our model, we evaluate on extrapolated viewpoints that fall outside the range of input views. This constitutes a more challenging setting for NVS, leading to increased uncertainty and unobserved regions. As summarized in Table 2, CoDiffSplat achieves

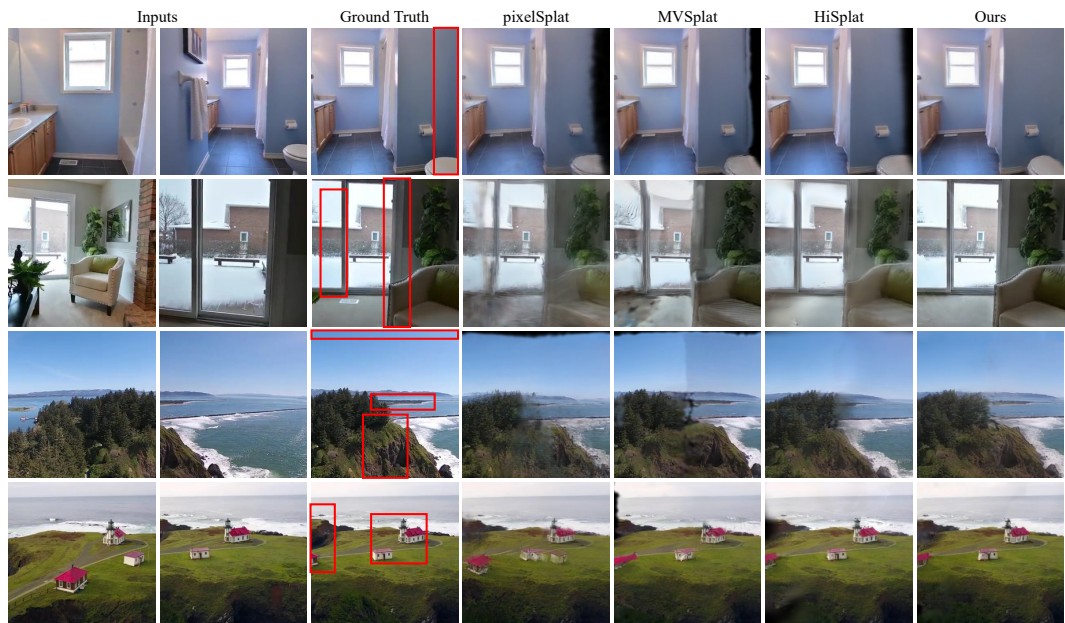

Figure 3: **Qualitative comparison of interpolated NVS.** The first two columns show sparse input views, while the third column presents the ground truth for the target interpolated view between them. CoDiffSplat better reconstructs fine details and occluded regions (highlighted in red) in both indoor (RealEstate10K, top two rows) and outdoor (ACID, bottom two rows) settings.

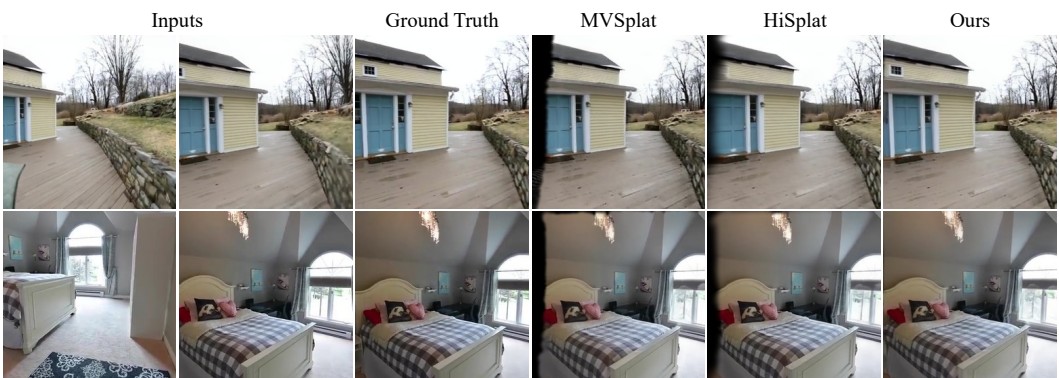

Figure 4: **Qualitative results for extrapolated NVS on RealEstate10K.** Baseline methods exhibit voids and distorted geometry in unseen regions, while CoDiffSplat reduces missing-geometry artifacts and better preserves boundary structures.

consistent improvements (+2.32 dB PSNR and 20% LPIPS reduction over the best baseline). As shown in Figure 4, pixel-aligned estimators (MVSplat and HiSplat) often produce "black holes" or geometric artifacts in unseen areas. In contrast, our method reduces missing-geometry artifacts and preserves fine structures. We attribute these gains to CEA conditioning and the single-step diffusion refinement, which regularizes Gaussian parameters with semantic cues beyond pixel-level correspondences.

## 4.3 ABLATION STUDIES

### 4.3.1 ABLATIONS ON DIFFUSION STRATEGY

We ablate our diffusion strategy from three aspects: (i) removing diffusion (*w/o Diffusion*), where the initial Gaussians are directly upsampled for rendering, (ii) replacing the DiT backbone with a

Table 3: **Ablations of CoDiffSplat.** Models trained on RealEstate10K with two input views. *w/o Diffusion* removes the diffusion process and renders directly from the initial Gaussians. *w/ UNet Backbone* substitutes the DiT with a 2D UNet. *Full (10 steps)* introduces multi-step diffusion.

| Setup | PSNR↑ | SSIM↑ | LPIPS↓ |
|---|---|---|---|
| w/o Diffusion | 25.47 (-2.09) | 0.849 (-0.039) | 0.149 (+0.035) |
| w/ UNet Backbone | 26.71 (-0.85) | 0.873 (-0.015) | 0.123 (+0.009) |
| Full (10 steps) | 27.05 (-0.51) | 0.876 (-0.012) | 0.118 (+0.004) |
| Full (1 step) | **27.56** | **0.888** | **0.114** |

Table 4: **Ablations on conditional embeddings.** Models trained on RealEstate10K with two input views. CEA outperforms DINOv3, CLIP, and BLIP-2 under identical conditioning interfaces and training settings. Green indicates drops relative to the best model (CEA).

| Setup | PSNR↑ | SSIM↑ | LPIPS↓ |
|---|---|---|---|
| Class Embedding | 26.13 (-1.43) | 0.859 (-0.029) | 0.139 (+0.025) |
| CLIP Embedding | 26.41 (-1.15) | 0.867 (-0.021) | 0.126 (+0.012) |
| BLIP Embedding | 26.54 (-1.02) | 0.870 (-0.018) | 0.125 (+0.011) |
| CEA Embedding | **27.56** | **0.888** | **0.114** |

2D UNet (*w/ UNet Backbone*), and (iii) varying the number of diffusion steps (*Full (10 steps)*). As summarized in Table 3, excluding diffusion significantly degrades performance, indicating its necessity for effective refinement. Replacing DiT with a UNet also lowers accuracy, which we attribute to the inductive bias of convolutional backbones that enforce 2D grid structures, unsuitable for unordered 3D Gaussian primitives. Moreover, while classical diffusion models typically rely on multi-step denoising, we find that single-step diffusion yields better balance of fidelity and stability, outperforming the 10-step variant by +0.5dB PSNR. We hypothesize that additional steps introduce stochastic perturbations that accumulate into over-smoothing and hallucinated structures. Further qualitative comparisons are provided in Appendix A.2.2.

### 4.3.2 ABLATIONS ON CONDITIONAL EMBEDDINGS

To isolate the effect of conditioning, we compare the proposed Cross-view Entropy-Aware (CEA) embedding with three common alternatives: (i) a class embedding from a DINOv3-pretrained ViT, (ii) a CLIP image–text embedding, and (iii) a BLIP-2 pseudo-caption embedding. All embeddings are used under identical training and injection settings. As shown in Table 4, CEA achieves consistent gains over the alternatives; for example, it improves PSNR by +1.02 dB over the next best variant (BLIP) and reduces LPIPS by ∼8.8%. We attribute this gap to CEA's cross-view fusion and entropy-aware weighting, which emphasize uncertain regions and thereby reduce missing-geometry artifacts and boundary erosion. For more qualitative discussions, please refer to the Appendix A.2.3.

## 5 CONCLUSION

In this work, we propose CoDiffSplat, a novel G-3DGS framework for novel view synthesis from sparse-view inputs. Unlike prior methods that rely on pixel-aligned Gaussian estimation, our approach leverages single-step conditional diffusion to refine the Gaussians, effectively compensating and refining geometry in partially observed or uncertain scenes. To enhance both global semantic coherence and fine-grained detail awareness, we introduce a Cross-View Entropy-Aware module that produces semantically rich embeddings for conditioning the diffusion. Extensive experiments on multiple datasets demonstrate that CoDiffSplat significantly enhances reconstruction fidelity, particularly in occluded and unseen regions. These results highlight the promising potential of diffusion-based paradigms in generalizable 3DGS. However, extreme extrapolation remains challenging, where geometric collapse can still occur. Furthermore, our current training is limited to RealEstate10K and ACID. Scaling to broader datasets or jointly training across diverse domains may unlock richer and more robust diffusion priors.

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
