# OpenReview forum: "CoDiffSplat: Sparse-View Generalizable 3D Gaussian Splatting with Single-Step Conditional Diffusion"
_ICLR.cc/2026/Conference — ICLR 2026 Conference Withdrawn Submission_

### Official Review · Reviewer_4TGp · 2025-10-28

**Soundness:** 3
**Presentation:** 3
**Contribution:** 2
**Rating:** 4
**Confidence:** 4

**Summary:**

This work focuses on the sparse-view 3DGS (more specifically, generalizable 3DGS) and points out the limitation of the "incomplete geometry and structural collapse" in the occluded regions. To this end, this work proposes CoDiffSplat framework, which combines the semantic-condition and 3D Gaussians using the diffusion model. Besides, this work adopts the lightweight single-step diffusion and Cross-View Entropy-Aware (CEA) for the pointed issue. Experimental results on different datasets show that CoDiffSplat achieves improvements compared to baselines.

**Strengths:**

* This work is well-written, and the readers can get the proposed points and understand the proposed pipeline straightly.
* The experiments on the RealEstate10K dataset show better results and steady improvements compared to HiSplat.

**Weaknesses:**

* The significance of this work.
  * The core of this work. In my opinion, the main problem addressed by this work is the "black hole" problem in the extrapolation scenario, as shown in its Introduction and experimental results. However, I think this problem may not be a particularly important problem, and it may even be very easy to solve in the field of generation. In Figures 3 and 4, we can notice that the main contribution of this work is filling the "black holes" on the edges, but we can also notice that the supplemented regions do not introduce too much additional visual information, but are rather "relatively simple filling". Meanwhile, we can also infer that the significant improvement in the metrics shown by extrapolated NVS on RealEstate10K (Table 2) is most likely from these "black holes" on the edges.
  * Are there similar or even better methods in other fields? In the field of video generation, camera-control video generation actually solves similar problems (e.g., TrajectoryCrafter, Reangle-A-Video). These models can not only achieve extrapolation of wild data (while this work is the result of the specific datasets), but also achieve extrapolation to a wider angle.
  * Change the idea to solve the sparse-view issues. Based on the above, I think that directly solving the sparse-view issue from the sparse viewpoint images is not the best approach. Instead, we should first use a generative model to augment the data, and then use the generalized 3DGS method to obtain an explicit 3DGS representation (of course, we can also improve the generative model and G-3DGS to make them more coupled). This may be a more important and reasonable approach.
  * Overall, I think the problem this work solves is of limited significance.

**Questions:**

* I am curious about the experimental results of ablations shown in Table 2. Why "Full (10 steps)" performs worse than "Full (1 step)"? Can you provide more detailed explanations and illustrations about this issue?

---

### Official Review · Reviewer_CFAR · 2025-10-30

**Soundness:** 2
**Presentation:** 3
**Contribution:** 3
**Rating:** 4
**Confidence:** 4

**Summary:**

This paper introduces CoDiffSplat, a novel framework for sparse-view generalizable 3D Gaussian splatting. Conditional diffusion is introduced to refine Gaussian parameters, where they employ a single-step diffusion process in latent Gaussian space, guided by a Cross-View Entropy-Aware (CEA) module to resolve ambiguities under occlusion and sparse overlap. Key contributions include: (1) They employ a lightweight DiT-based diffusion model with a single-step denoiser to relax rigid pixel alignment and restore missing geometry, (2) A CEA module is proposed to provide fine-grained conditional guidance for diffusion, and (3) Experiments on RealEstate10K and ACID show gains over current leading methods.

**Strengths:**

1.Unlike conventional pixel-aligned paradigms, the work introduces conditional diffusion to achieve better 3D Gaussian reconstruction. The single-step diffusion formulation for feedforward 3D Gaussian splatting is novel, as prior diffusion-based 3D methods focus more on full generative pipelines rather than efficient correction modules. The CEA module introduces a unique entropy-aware mechanism to fuse cross-view semantics, trying to address occlusion and ambiguity in their reconstruction process.

2.​​Experiments on RealEstate10K and ACID demonstrate consistent improvements in PSNR, SSIM, and LPIPS over strong baselines (e.g., HiSplat, MVSplat). Ablation studies validate some of the design choices, such as the superiority of DiT over UNet backbones and the effectiveness of single-step diffusion.

3.​​The paper is basically well-written (some issues still exist, see weakness) with clear motivations and method explanations on most components.

**Weaknesses:**

1.Diffusion-based sparse view reconstruction methods have been explored under the Nerf paradigm. Works like [1] or [2] are not sufficiently discussed in the current version. Similarly​, the comparative experiments lack diffusion-based method. Authors should consider comparing experiments on works like [1] as mentioned.

2.The explanation regarding the single-step refinement (3.2.3) is somewhat inadequate. The details of their utilized DIT architecture is not sufficiently discussed (already check the appendix, not provided). Since [3] propose several variants of DIT, it is unclear which one they use. Further, the intuition behind adopting DiT is not sufficiently discussed. The authors may consider provide more discussions.

3.To validate the efficiency of their methods, they conduct experiments on the inference time. My concern is the computational parameters​ for their methods, since both ViT and DiT architecture is used in their method. Authors may consider providing experiments on the param.

4.The ablation study on the diffusion process appears somewhat limited, particularly regarding the choice of the number of diffusion steps (only 1 and 10). Wider range of steps should be considered to strengthen their point that “a single-step refinement suffices to correct geometric inconsistencies, turning diffusion into an efficient correction module rather than a full generative process”

5.The ablation study of CEA module is insufficient. The authors utilize different embedding methods to conduct the experiments. They should also consider removing the CEA to achieve a more comprehensive ablation analysis.

Reference:

[1] Wu, R., Mildenhall, B., Henzler, P., Park, K., Gao, R., Watson, D., Srinivasan, P.P., Verbin, D., Barron, J.T., Poole, B., & Holynski, A. (2023). ReconFusion: 3D Reconstruction with Diffusion Priors. 2024 IEEE/CVF Conference on Computer Vision and Pattern Recognition (CVPR), 21551-21561.

[2] Zhou, Z., & Tulsiani, S. (2022). SparseFusion: Distilling View-Conditioned Diffusion for 3D Reconstruction. 2023 IEEE/CVF Conference on Computer Vision and Pattern Recognition (CVPR), 12588-12597.

[3] Peebles, W.S., & Xie, S. (2022). Scalable Diffusion Models with Transformers. 2023 IEEE/CVF International Conference on Computer Vision (ICCV), 4172-4182.

**Questions:**

1.What is the intuition of utilizing a DiT? (See weakness 2 for details)

2.In Section3.3, they use a factor 0.05 for their LPIPS part of their overall loss, what is the reason behind this? Are there any experiments or reference for the choice of the factor?

For other questions, please see the weakness. I will consider raising my ratings if my concerns are well addressed.

---

### Official Review · Reviewer_D7Dd · 2025-11-03

**Soundness:** 2
**Presentation:** 3
**Contribution:** 1
**Rating:** 2
**Confidence:** 4

**Summary:**

This work proposes introducing semantic-conditioned latent diffusion to enhance the reconstruction performance of Generalizable 3D Gaussian Splatting under sparse views. The authors claim that this method achieves better performance compared to previous works.

**Strengths:**

1. This work is well-written, especially in the description of the method.

2. Experiments demonstrate that the proposed method achieves improvements over previous approaches, particularly in the extrapolated NVS task.

**Weaknesses:**

1. The novelty of this work is limited. Firstly, introducing diffusion models to refine 3D Gaussian Splatting under sparse view settings is a common approach [1]. Secondly, the initialization process in Section 3.1 is similar to MVSplat [2], while the conditional diffusion in Section 3.2 resembles some existing multi-view generation works [3].

[1] 3dgs-enhancer: Enhancing unbounded 3d gaussian splatting with view-consistent 2d diffusion priors.

[2] Mvsplat: Efficient 3d gaussian splatting from sparse multi-view images.

[3] Shape from Semantics: 3D Shape Generation from Multi-View Semantics

2. The diffusion lacks control over detailed aspects. The paper introduces DINOv3 as a condition to guide the diffusion process, which is often a high-level condition. Theoretically, it lacks the ability to control the generation of fine details, especially in high-frequency regions.

3. In cross-dataset validation, the performance improvement brought by CoDiffSplat is noticeably limited, which may be due to the limited generalization capability of the diffusion process. Moreover, this part of the experiment is crucial for G-3DGS and should have been included in the main text.

4. Compared to previous works, the introduction of an additional diffusion model results in a significant decrease in efficiency for CoDiffSplat compared to MVSplat.

5. The paper contains some formatting errors, such as issues with citation formatting.

**Questions:**

1. What are the experimental setup details for the Extrapolated NVS task?

2. How does CoDiffSplat perform with more input views? Does it require retraining for different numbers of inputs?

---

### Official Review · Reviewer_nVFc · 2025-11-03

**Soundness:** 3
**Presentation:** 3
**Contribution:** 3
**Rating:** 4
**Confidence:** 4

**Summary:**

This paper aims to address the quality degradation of G-3DGS reconstruction under sparse-view settings. The paper attributes the problem to the pixel-level alignment strategy, which is overly sensitive to depth estimation quality and hinders the reconstruction of partially visible regions. To address the problem, this paper propose a diffusion-based framework called CoDiffSplat to refine pixel-aligned Gaussians in only one step. This paper further proposes a CEA module to inject semantic information of multi-views into diffusion models as conditions.

**Strengths:**

1. The refinement pipeline in Gaussian Space provides better consistency between multi-views.
2. This paper is well-organized and easy to read.

**Weaknesses:**

1. The main contribution of this paper lies in proposing a one-step Gaussian refinement diffusion model. However, in Sec. 4.2, the comparisons are limited to reconstruction algorithms. It is necessary to further include comparisons with refinement methods (e.g., DIFIX3D+) to better demonstrate the paper’s contribution.
2. In Table3, why does multi-step denoising lead to a significant performance degradation compared with the single-step variant? Does this imply that a transformer-based feed-forward network can already achieve competitive performance for Gaussian refinement? The authors may need to conduct experiments with a transformer-based feed-forward baseline to justify and explain the necessity of diffusion modeling.

**Questions:**

Please see the weakness.

---

### Note · Authors · 2025-11-12

I have read and agree with the venue's withdrawal policy on behalf of myself and my co-authors.